# *Only Words Count; the Rest Is Mere Chattering*: A Cross-Disciplinary Approach to the Verbal Expression of Emotional Experience

**DOI:** 10.3390/bs12080292

**Published:** 2022-08-18

**Authors:** Daniela Laricchiuta, Andrea Termine, Carlo Fabrizio, Noemi Passarello, Francesca Greco, Fabrizio Piras, Eleonora Picerni, Debora Cutuli, Andrea Marini, Laura Mandolesi, Gianfranco Spalletta, Laura Petrosini

**Affiliations:** 1IRCCS Fondazione Santa Lucia, 00143 Rome, Italy; 2Department of Humanities, Federico II University of Naples, 80138 Naples, Italy; 3Department of Communication and Social Research, Sapienza University of Rome, 00198 Rome, Italy; 4Department of Psychology, University “Sapienza” of Rome, 00185 Rome, Italy; 5Department of Languages, Literatures, Communication, Education and Society, University of Udine, 33100 Udine, Italy

**Keywords:** alexithymia, emotional processing, cortical thickness, Natural Language Processing, machine learning

## Abstract

The analysis of sequences of words and prosody, meter, and rhythm provided in an interview addressing the capacity to identify and describe emotions represents a powerful tool to reveal emotional processing. The ability to express and identify emotions was analyzed by means of the Toronto Structured Interview for Alexithymia (TSIA), and TSIA transcripts were analyzed by Natural Language Processing to shed light on verbal features. The brain correlates of the capacity to translate emotional experience into words were determined through cortical thickness measures. A machine learning methodology proved that individuals with deficits in identifying and describing emotions (n = 7) produced language distortions, frequently used the present tense of auxiliary verbs, and few possessive determiners, as well as scarcely connected the speech, in comparison to individuals without deficits (n = 7). Interestingly, they showed high cortical thickness at left temporal pole and low at isthmus of the right cingulate cortex. Overall, we identified the neuro-linguistic pattern of the expression of emotional experience.

## 1. Introduction

The representations of each word we use are maintained in a mental lexicon containing information about pronunciation, syntax, general meaning, and affective connotations [1]. Studies on the language of emotion have focused on affective words, and influential theories have assigned to language a role in the development of emotions, associating language deficits with impaired emotional processing [2].

Deepening the scope of emotional processing, the Multiple Code Theory posits that emotional information is represented in verbal symbolic and subsymbolic systems as well as nonverbal symbolic and subsymbolic systems [3,4]. Of these, the verbal symbolic system refers to the capacity of language to direct and regulate action and to activate imagination and emotion. It concerns those aspects of mental functions that pertain to words as discrete units of meaning that are strung together to form increasingly complex representations. The verbal subsymbolic system concerns aspects such as prosody, meter, rhythm, and phonemic qualities of language. The two channels of the nonverbal system include images (symbolic system) and proceedings (subsymbolic system) related to implicit elaboration associated with visceral, somatic, sensory, and motor modalities. Translating emotional experience into words corresponds to the transition from the subsymbolic (i.e., implicit) level to the symbolic (i.e., explicit) level [3]. Such a computation contributes to interpreting visceral states related to emotional expression and to distinguishing them from physiological states [4].

Contemporary neuroscience recognizes that emotional expression and emotional experience are characterized by incompletely overlapping biological correlates distributed across brain circuits and bodily organs [5]. Specifically, emotional expression occurs at a physiological and motor-expressive level, and involves changes in visceral states and somatosensory cortical and limbic areas [6]. Emotional experience is based on complex symbolizations and explicit processes related to the functioning of the prefrontal, temporal, and parietal cortices. The integration of emotional expression and emotional experience would depend on cooperation between cortical and limbic structures.

It is reasonable to posit that the translation of emotional experience into words is associated with language style, since narratives thoroughly mirror emotional processes [7] (Ionesco once said that “only words count and the rest is mere chattering”). Consequently, the verbal symbolic and subsymbolic analysis of narratives provided in an interview addressing the capacity to identify and describe emotions is a powerful tool to reveal emotional processing. The present study used this tool. Namely, transcripts and speech recordings obtained through the Toronto Structured Interview for Alexithymia (TSIA) [8] were analyzed to shed light on verbal symbolic and subsymbolic features linked to the capacity to identify and describe emotions. In fact, the TSIA represents a validated method to evaluate the level of alexithymia (“no words for emotion”), a personality construct characterized by marked difficulties in identifying emotions, finding appropriate words to describe them, and distinguishing feelings from bodily sensations [8]. Importantly, alexithymia can result from a failure of emotional categorization [9,10] or from a lack of relocation from the embodied internal state to the symbolic representation of emotional signals [3,11].

On this basis, the current study recruited a sample of healthy subjects and analyzed their ability to regulate, express, and identify emotions by means of the TSIA [8,12], Attachment Style Questionnaire (ASQ) [13], Emotion Regulation Questionnaire (ERQ) [14], Coping Orientation to Problems Experiences (COPE) [15], Beck Depression Inventory (BDI) [11,16], and State–Trait Anxiety Inventory-2 (STAI-2) [11,17]. Cortical thickness was also measured by 3-T high-resolution structural magnetic resonance imaging (MRI) to analyze the cortical correlates of the capacity to translate emotional experiences into words. The above-described TSIA transcripts were analyzed by using the Natural Language Processing (NLP) techniques [18,19,20]. Furthermore, TSIA recordings were analyzed by using OpenSMILE API [21] to extract acoustic features.

Through a forward-looking machine learning methodology, cross-disciplinary (linguistic, acoustic, neuroimaging, and psychological) outcomes were combined to obtain a holistic representation of the elements that contribute to the expression of emotional experience.

## 2. Materials and Methods

### 2.1. Ethics Statement

The study was approved by the Local Ethics Committee of the IRCCS Fondazione Santa Lucia, Rome, and written consent was obtained from all participants after the study procedures were fully explained to them.

### 2.2. Participants 

Starting from an a priori power analysis to compute the sample size (please see the 2.3 Transparency and Openness paragraph), we included in the present study 22 subjects ((12 males); aged 21–63 years (mean 44.32 ± 13.17); mean years of formal education 15.68 ± 2.40)) belonging to a larger group (n = 79) of healthy volunteers submitted to MRI scan protocol for other studies at the IRCCS Fondazione Santa Lucia, Rome, under typical exclusion and inclusion criteria detailed elsewhere [11].

Briefly, inclusion criteria were: age between 18 and 65 years and suitability for MRI scanning. Exclusion criteria included: (1) cognitive impairment or dementia and confirmed by clinical neuropsychological evaluation; (2) subjective complaint of memory difficulties or of any other cognitive deficit, regardless of interference with daily activities; (3) major medical illnesses, e.g., diabetes (not stabilized), obstructive pulmonary disease, or asthma; hematologic and oncologic disorders; pernicious anemia; clinically significant gastrointestinal, renal, hepatic, endocrine, or cardio-vascular system diseases; newly treated hypothyroidism; (4) current or reported mental or neurological disorders (e.g., schizophrenia, mood disorders, anxiety disorders, stroke, Parkinson’s disease, seizure disorder, head injury with loss of consciousness, and any other significant mental or neurological disorder); (5) known or suspected history of alcoholism or drug dependence and abuse; (6) MRI evidence of focal parenchymal abnormalities or cerebro-vascular diseases; for each subject, a trained neuroradiologist and a neuropsychologist expert in neuroimaging co-inspected all the available clinical MRI sequences (i.e., T1- and T2-weighted and FLAIR images) to ensure that the subjects were free from structural brain pathologies and vascular lesions (i.e., FLAIR or T2-weighted hyper-intensities and T1-weighted hypo-intensities).

### 2.3. Transparency and Openness

In the present research, three groups were used, defined by total TSIA scores: subjects with scores ≤13 are classified as Low Alexithymia (LA) group, those with scores between 13 and 39 are classified as Medium Alexithymia (MA) group, and those with scores ≥39 are classified as High Alexithymia (HA) group. These thresholds represent 0.25 and 0.75 quantiles of the TSIA distribution. The sample size for hypothesis testing was computed using an a priori power analysis for the Wilcoxon rank-sum test for independent groups (LA and HA groups). Power analysis was performed on one of the most statistically significant parameters from linguistic profiling analysis, namely the distribution of possessive determiners in the groups (LA group: mean 0.98, standard deviation 0.18; HA group: mean 0.67, standard deviation 0.14). The sample size required was estimated in LA subjects n = 7, and HA subjects n = 7, with α err prob = 0.05, Effect size d = 1.92, achieved statistical power = 0.95, and the data used in this analysis were obtained from the sample included in this experiment. Required sample size and achieved statistical power were obtained using G*power v. 3.1.9.7. Thus, we used LA group (n = 7), MA group (n = 8), and HA group (n = 7).

The datasets generated during and/or analyzed during the current study are available from the corresponding author on reasonable request. This study’s design and its analysis were not pre-registered but performed following the STROBE statement that developed recommendations on what should be included in an accurate and complete report of an observational study.

### 2.4. Psychological Assessment

Participants underwent psychological assessment by means of the TSIA, ASQ, ERQ, COPE, BDI, and STAI-2. Italian versions of the psychological scales were used.

#### 2.4.1. TSIA Evaluation

The participants were interviewed through the TSIA, composed of 24 items [8,12] relating to the four factors of alexithymia construct. Each item was assessed by a specific open-ended question and responses were 3-point (coded ‘0’, ‘1’, or ‘2’) scored. The sum of scores resulted in a total score ranging from 0 (low alexithymia levels) to 48 (high alexithymia levels). For each item, the subject is asked to give one or more examples to illustrate and to support the answer. The scoring for each question is tied to the specific content of these examples by an expert interviewer and in general a score of ‘0’ is assigned if a specific difficulty is never or rarely present; ‘1’ if it is present sometimes; and ‘2’ if it is present many times. Cut-off scores have not been established. 

#### 2.4.2. ASQ Evaluation

Attachment styles were assessed through ASQ [13]. The 40-item self-report scale ASQ yields scores on five scales or dimensions: Confidence, Discomfort with Closeness, Relationships as Secondary, Need for Approval, and Preoccupation with Relationships. ASQ consists of propositions related to various aspects of attachment. On a 6-point Likert-type scale from totally disagree (1) to totally agree (6), the participants indicated the degree to which the propositions described their feelings.

#### 2.4.3. ERQ Evaluation

The ERQ [14] is a 10-item, self-report measure of two emotional regulation strategies of dealing with emotional arousal, one considered beneficial and the other harmful. Cognitive Reappraisal (measured by six items) is an adaptive, antecedent-focused strategy, affecting the early cognitive stages of emotional activity, whereby the initial interpretation of a given situation is re-evaluated. In contrast, Expressive Suppression (measured by four items) is a maladaptive, response-focused plan of action implemented after an emotional response has fully developed; it is conceptualized as inhibiting the behavioral expression of the emotion. The ERQ uses a 7-point Likert scale ranging from 1 (strongly disagree) to 7 (strongly agree). 

#### 2.4.4. COPE Evaluation

The COPE [15] is a multidimensional coping questionnaire that assesses coping strategies in response to stressful events. It consists of fifteen four-item scales designed to measure five dimensions of coping strategy: Seeking Social Support, Avoidance, Positive Attitude, Problem Solving, and Turning to Religion. Each item is rated on a 4-point Likert scale ranging from “Never” (1) to “Often” (4; scores ranging from 4 to 16 and higher scores representing higher frequency of coping strategies).

#### 2.4.5. BDI Evaluation

Depression levels were assessed through the BDI [11,16]. It is a 21-item multiple-choice self-report inventory rated on a four-point scale ranging from 0 to 3 that assesses the presence of depressive symptoms. Scores can range from 0 to 63. The questionnaire is composed of items relating to symptoms of depression such as hopelessness and irritability, cognitions such as guilt or feelings of being punished, as well as physical symptoms such as fatigue, weight loss, and lack of interest in sex.

#### 2.4.6. STAI-2 Evaluation

Trait anxiety levels were assessed through the scale 2 of STAI [11,17], a scale with 20 items rated on a four-point scale ranging from 1 to 4 assessing trait anxiety. Scores can range from 20 to 80.

### 2.5. Statistical Analyses for Socio-Demographic and Psychological Variables

TSIA score differences were assessed by unpaired Wilcoxon rank-sum test between females and males. Not parametric associations (Spearman, Confidence Interval 99%, corrected for multiple testing using False Discovery Rate) were analyzed between TSIA scores and age and formal education years. Furthermore, in LA, MA, and HA subject differences in age and years of formal education were analyzed by using Kruskal-Wallis. The differences were considered significant at the *p* ≤ 0.05 level. Data were analyzed using GraphPad Prism 5.03 (GraphPad Software, Inc., San Diego, CA, USA).

### 2.6. NLP Analysis

TSIA transcripts were preprocessed using Universal Dependencies (UD) Pipe [22]. This state-of-the-art linguistic annotation pipeline performs the following steps: sentence splitting, tokenization, part-of-speech (PoS) tagging, lemmatization, and dependency parsing [18,22]. Linguistic profiling was carried out using Profiling-UD [23]. Profiling-UD enables the extraction of more than 130 features spanning different levels of linguistic description by performing PoS tagging according to multilingual standardization criteria and the taxonomy of the UD framework [22]. Lexical diversity and readability assessment analyses were performed using textstat_lexdiv and textstat_readability functions in the R library quanteda.textstats [18]. To select the variables potentially associated with TSIA scores, all features extracted in the linguistic profiling were tested by univariate correlation filtering using the Spearman method. The correlated set of features was compared between groups using multiple Wilcoxon tests, and *p* values were corrected by false discovery rate.

Semantic profiling was carried out starting from a keyword analysis based on token frequencies to identify potential semantic differences between groups using the χ2 test. Keywords related to the expression of emotions were highlighted using target word collocation analysis with a seed composed of the lemmatized forms of “sentire” and “provare” (English translation: to feel). We selected two token objects for words inside and outside of the 10-word windows containing the keywords, and then we computed the association of words with the keywords using the χ2 test [18]. Differences in the number of keyword-associated words were tested by a two-sample test for equality of proportions. Latent Semantic Analysis (LSA) was performed to identify latent semantic structural differences between groups [24]. Topics were identified using an unsupervised document classification technique, namely, Latent Dirichlet Allocation (LDA) [18,24], and topic frequencies were compared between groups using Wilcoxon rank sum test. To further inspect the most discriminatory groups of semantically related words, we defined three different dictionaries, labeled “body”, “family”, and “feeling”, and we performed a seeded LDA (s-LDA). Sentiment Analysis was performed to unveil the prevalent sentiment on a continuum from negative to positive polarity using the Open Polarity Enhanced Name Entity Recognition (OpeNER) project database (OpeNER, 2013). Sentiment scores were compared between groups using Wilcoxon rank-sum tests. Emotion Recognition Analysis was used to identify explicit and implicit emotions using the Canada National Research Council (NRC) Word–Emotion Association Lexicon, which is the most comprehensive lexicon ever built, with over 100 language dictionaries leveraging Google translate AI [25]. Frequencies of emotion-associated tokens were compared between groups using Wilcoxon rank sum tests.

### 2.7. MRI Acquisition and Processing

Participants underwent an imaging protocol including standard clinical sequences (FLAIR, DP-T2-weighted) and a volumetric whole-brain 3D high-resolution T1-weighted sequence, performed with a 3 T Allegra MR imager (Siemens, Erlangen, Germany), with a standard quadrature head coil. Volumetric whole-brain T1-weighted images were obtained in the sagittal plane using a Modified Driven Equilibrium Fourier Transform (MDEFT) sequence (Echo Time/Repetition Time -TE/TR- = 2.4/7.92 ms, flip angle 15°, voxel size 1 × 1 × 1 mm^3^). All planar sequence acquisitions were obtained in the plane of the anterior-posterior commissure line.

The FreeSurfer imaging analysis suite (v5.1.3, http://surfer.nmr.mgh.harvard.edu/, accessed on 26 January 2020) was used for cortical reconstruction of the whole brain [26]. With this software, the T1-weighted images were registered to the Talairach space of each participant’s brain with the skulls stripped. Images were then segmented into WM/GM tissue based on local and neighboring intensities. The cortical surface of each hemisphere was inflated to an average spherical surface to locate both the pial surface and the WM/GM boundary. Preprocessed images were visually inspected before including into subsequent statistical analyses. Any topological defects were excluded from the subsequent analyses. Cortical thickness was measured based on the shortest distance between the pial surface and the GM/WM boundary at each point across the cortical mantle. The regional thickness value at each vertex for each participant, was mapped to the surface of an average spherical surface using automated parcellation in FreeSurfer [27]. Segmentations of 68 (34 left and 34 right) cortical gray matter regions based on the Desikan–Killiany atlas [28] and two whole-hemisphere measures were visually inspected and statistically evaluated for outliers.

### 2.8. Acoustic Features

TSIA recordings were sampled to extract 10-sec samples from four interview items (one for each TSIA factor) for each subject using Audacity (http://audacityteam.org/, accessed on 26 January 2020). Automatic acoustic feature extraction was performed using the OpenSMILE API (v 2.0.2) in Python (v. 3.9.4) (Wilmington, DE, USA) with the 2-channel setting (Eyben et al., 2010). The ComParE feature set was computed from low-level descriptors using functional level setting [29]. Univariate filtering with Spearman correlation was applied to select a set of features associated with TSIA scores prior to machine learning modeling. Log-frequency power spectrograms were generated using the Librosa pipeline [30].

### 2.9. Machine Learning Integration of Cross-Disciplinary Data

Machine learning algorithms were used to predict TSIA scores exploiting two different datasets: one made of integrated NLP, psychological, neuroimaging and demographic data and the other made of audio features extracted from OpenSMILE application, including demographics. To avoid overfitting due to the high dimensionality of the datasets, appropriate machine learning methods were used, namely, elastic net (EN) regression, principal components regression (PCR), and partial least squares regression (PLSR), and their results were compared to find the best fit. EN is a machine learning method that implements implicit feature selection through a regularization approach combining ridge and lasso regression to penalize complex models by modulating regression coefficients [25]. PCR is a regression analysis based on Principal Components Analysis (PCA), and it was used as a regularized procedure, as in this type of analysis, the principal components of the explanatory variables are used as regressors [25]. Finally, PLSR was included as a technique similar to PCR, as it finds latent variables that maximally explain the relation between predictors and response features [25]. The leave-one-out cross-validation (LOOCV) technique was used for model training and testing to manage sample sizes. LOOCV uses n-1 samples for training, where n = 14 (n_HA_ = 7, n_LA_ = 7), and test statistics are computed on the left-out sample. The final models were selected by evaluating the R2 and root mean square error (RMSE) metrics of the models during cross-validation on both datasets. The importance of each feature in predicting TSIA scores was investigated by evaluating standardized regression coefficients.

### 2.10. Replicability and Validation Strategies

We decided to apply a robust data analysis workflow to provide replicable and reliable results. We determined the appropriate sample size for reliable hypothesis testing based on the a priori power analysis reported above. Linguistic profiling and emotion recognition were performed using the most widespread standardized reference corpus and pipeline to ensure replicability [22,25]. To avoid overfitting due to sample size and high dimensionality in our data we performed a two-stage feature selection process including dimensionality reduction, allowing us to summarize variance in the data in a few features (5 or 3 components), thus counteracting potential biases. Subsequently, we divided the machine learning analysis in two datasets, one for the acoustics features and one for NLP, neuropsychological, neuroimaging, and demographic variables. Results from machine learning modeling were validated with LOOCV technique, which allowed us to better assess generalizability while producing stable findings. To ensure results replicability, a seed was set to 12345 as guidance for algorithms needing pseudorandomization and R code is available upon reasonable request.

## 3. Results

### 3.1. Sociodemographic and Psychological Variables

Appendix A shows the participants’ demographic variables and scores on psychological scales (TSIA (total and factors: Difficulty in Identifying Feelings—DIF; Difficulty in Describing Feelings—DDF; Externally Oriented Thinking—EOT; and Imaginal Processes—IP); ASQ; ERQ; COPE; BDI; STAI-2) in the sample at large and in three groups.

No correlation emerged between TSIA scores (total or factors) and age or years of formal education. Furthermore, we found that LA, MA, and HA subjects did not differ in age (Kruskal–Wallis test P = 0.37) or years of formal education (Kruskal–Wallis test P = 0.36). These insignificant results in such important indicators as gender and education can probably be ascribed to the small number of participants in the samples.

The Wilcoxon rank-sum test revealed that males scored higher than females on the overall TSIA and in all TSIA factors except IP (at least U = 25; P = 0.02).

All indices (r and P adjusted) are reported in Table 1. The correlations were corrected for multiple testing using False Discovery Rate.

Indices (r and P adjusted) of correlation between Toronto Structured Interview for Alexithymia (TSIA) scores (total and factors: Difficulty in Identifying Feelings—DIF; Difficulty in Describing Feelings—DDF; Externally Oriented Thinking—EOT; and Imaginal Processes—IP) and age and years of formal education.

### 3.2. NLP Analysis

We compared transcripts from the three groups; the linguistic analysis of the extreme groups (HA and LA subjects) is reported in the main text, given that it best illustrates the linguistic differences associated with the degree of alexithymia. For the sake of completeness, comparisons of HA and LA subjects with MA subjects are reported in Appendix A.

#### 3.2.1. Linguistic Profiling, Linguistic Complexity, and Readability Assessment

We compared the transcripts from the three groups based on 130 indices derived from the UD framework. Overall, HA subjects used significantly (P adjusted = 0.03) more pseudowords or nonwords (language distortions, such as words that, for whatever reason, cannot be assigned a real part-of-speech category, e.g., mmmxh; tag: Other) (Figure 1(A1)), more (P adjusted = 0.03) present-tense forms of auxiliary verbs (Figure 1(A2)), and fewer possessive determiners (P adjusted = 0.01) (Figure 1(A3)) than LA subjects. Moreover, HA subjects had a less connected pattern of speech, as demonstrated by the differences in the use of words mapped onto coordinating conjunctions (e.g., both … and) (P adjusted = 0.02) (Figure 1(A4)) and coordinating conjunction relation (the relation between a sentence-initial coordinating conjunction and the sentence root) (P adjusted = 0.03) (Figure 1(A5)). No significant difference was found between HA and LA subjects in linguistic complexity (16 indices) or text readability (48 indices). Comparisons of HA and LA subjects with MA subjects are reported in Appendix A.

#### 3.2.2. Semantics: Keyword Assessment and Topic Modeling

When the keyword frequencies of HA and LA subjects were compared using the χ2 test, 222 keywords were found to be significant. HA subjects used fewer emotional keywords (such as “feeling”, “happy”, and “to express”) and more filler words (“maybe”, “in conclusion”) than LA subjects (Figure 1B). This differential use of keywords was confirmed by unsupervised LSA, which showed that these two groups formed separate clusters in a two-dimensional space representing the latent semantic structure. The orthogonal dimensions represent semantically related clusters of keywords, and LA and HA subjects were divided by Dimensions 4 and 5 in the LSA space (Figure 1C).

The relation between alexithymia and keywords related to emotional expression was detailed by Target Word Collocation Analysis. We chose the emotional Italian words “sentire” and “provare” (both translated in English as to feel), which were in the top five differentially used words and could represent important keywords to establish a network. The resulting network was smaller in HA (P < 0.001) than in LA subjects (Appendix A) (comparisons between HA or LA and MA subjects are shown in Appendix A).

Since differential keyword patterns may be related to deep semantic differences, we performed Topic Modeling. To assess how the different groups of words were organized in topics, we performed unsupervised LDA and identified 20 topics, described by the top five words by frequency of use. Three topics were significantly different between HA and LA subjects (Wilcoxon rank sum test: P ≤ 0.05; Topic 1: fashion, reason, end, minute, effect; Topic 2: day, home, week, sister, vacation; Topic 3: sense, parent, news, scene, patient) (Figure 1D). A further description of topics was performed by s-LDA mapping subjects’ narratives to predefined topics. Specifically, we built three dictionaries containing words related to bodily parts or sensations (“body”), relational words (“family”), or emotional words (“feeling”). HA subjects showed elevated frequencies of “body”-related words and no “feeling”-related words (Appendix A). Comparisons of HA and LA subjects with MA subjects in terms of keyword assessment and topic modeling results are reported in Appendix A.

#### 3.2.3. Sentiment Analysis and Emotion Recognition Analysis

To explore sentiment polarization, we applied Sentiment Analysis, which enabled us to evaluate the subjective information in an expression. HA and LA subjects showed different sentiment scores (Wilcoxon rank sum test: P = 0.026), with HA subjects being more neutral in the continuum between positive and negative sentiments. To inspect which emotions were expressed, the Emotion Recognition Analysis was performed by mapping all lemmas of the transcripts onto a database of emotion-related words. HA subjects displayed a reduced frequency of emotion-related words specifically for anger (P = 0.025) and disgust (P = 0.024) compared to LA subjects (Figure 1E) (comparisons of HA and LA subjects with MA subjects are presented in Appendix A).

### 3.3. Machine Learning Analysis of Cross-Disciplinary Data

NLP data were aggregated with demographic, neuroimaging, and psychological data to predict total TSIA scores. Candidate features (n = 92) were extracted using univariate filtering with Spearman correlation. After the data were fitted with EN regression, PCR, and PLSR, a PCR model with three components was selected, as it had the best performance in the evaluation metrics assessed by the LOOCV (R^2^ = 0.86; RMSE = 0.35) (Figure 2A,B). Variable importance is reported in Figure 2(C1,C2). Of the 92 candidate features, 85 were derived from NLP data, 3 from psychological measures, 2 from neuroimaging data, and 2 (gender and age) from demographic data. The variable importance of NLP linguistic profiling indices confirmed syntactical and semantic features associated with alexithymia. For example, while frequent use of present tenses predicted high TSIA scores, frequent use of past tenses and speech connectivity predicted low TSIA scores. Frequent use of words associated with emotions such as anger, disgust, and happiness was associated with low TSIA scores, while frequent use of pseudowords and concrete words was associated with high TSIA scores.

Regarding the variable importance of psychological measures, high ERQ Expressive Suppression and ASQ Discomfort with Intimacy scores predicted high TSIA scores. Conversely, high ASQ Confidence scores predicted low alexithymia levels.

Regarding the variable importance of neuroimaging data, increased cortical thickness at the left temporal pole (Figure 2(D1)) predicted high TSIA scores, while decreased cortical thickness at the isthmus of the right cingulate cortex (Figure 2(D2)) predicted low TSIA scores.

### 3.4. Machine Learning Analysis of Speech Records

Univariate correlation filtering was applied to select 212 candidate acoustic features belonging to the ComParE feature set. A PCR model with five components was selected as the best-fitting model based on the R2 and RMSE metrics from LOOCV (R^2^ = 0.74; RMSE = 6.74) (Figure 3A,B). Relative variable importance was computed over the domains of low-level descriptors in the ComParE feature set, namely, sound quality as well as spectral, cepstral and prosodic features (Figure 3C). Notably, prosodic and spectral acoustic features were the most relevant for TSIA score prediction. Variable importance is reported in detail in Appendix A. 

Prosodic features are perceptible changes in speech rhythm, stress, and intonation over long segments of time. Among these features, voice signal (fundamental frequency—F0) variation was negatively associated with TSIA scores. Moreover, normalized length of Relative Spectra (RASTA), which represents a correction factor for nonlinguistic components in speech, was found to be positively associated with TSIA scores, implying different voice quality in the speech of HA subjects.

Among the spectral features of speech, the roll-off frequency can be used to distinguish between harmonic and noisy sounds. Harmonicity variation in roll-off frequency was negatively associated with TSIA scores. Cepstral descriptors are “filter features” describing the resonant properties of the vocal and nasal tract filter. Mel-Frequency Cepstral Coefficients (MFCCs) represent the envelope of the short-term power spectrum, which is a manifestation of the shape of the vocal tract. Coefficients ranging from 0–14 bands represent the peaks’ Simple Moving Average (SMA) low-pass filtering in voiced regions. SMA over peak frequency distribution was positively associated with TSIA scores, while Linear Predictive Coding (LPC), representing envelopments in vocal tract shape, was negatively associated with TSIA scores. Among sound quality descriptors, jitter, indicating an irregularity in voice intensity, and Harmonics-to-Noise-Ratio (HNR), a measure of voice quality, were positively and negatively associated with TSIA scores, respectively. Representative differences in audio tracks of HA and LA subjects are reported in Figure 3(D1,D2).

## 4. Discussion

Since the way language is used to describe emotion indicates what one knows about emotion and how emotion is experienced [1], individuals with high levels of alexithymic traits may be unable to form an accessible verbal representation of their emotional state [31]. Surprisingly, however, the relation between alexithymia and language has been understudied. While most studies have counted the occurrence of emotional words, only a few studies have examined verbal encodings of emotion as a means of direct access to the symbolization of emotion [7,9,10].

Specifically, people with high alexithymia use uncomplicated emotional language [10], and ectrophysiologically, they show low sensitivity to the emotional speech [32], suggesting that alexithymia affects the way qualities of emotional lexicon are processed by the brain.

In the current innovative study, by using NPL and acoustic feature analyses, we detailed the verbal symbolic and subsymbolic means used to refer to emotions by individuals with different levels of alexithymia. This investigation was combined with psychological and neuroimaging analyses. Several replicability and validation strategies were applied to guide study design and linguistic reference mapping [22]. Feature selection and dimensionality reduction techniques, along with models independent testing, were used in the analysis pipeline to ensure reliable findings, following best data practices for data mining. 

In accordance with previous findings [11], males showed higher alexithymia levels than females. The relationship between alexithymia and attachment styles indicated that people with high alexithymia were less confident and more uncomfortable with closeness and relationships than people with low alexithymia. Furthermore, people with high alexithymia tended to inhibit emotional expression, adopting maladaptive response-focused behaviors. Previous psychodynamic theories characterized alexithymia as closely related to psychological trauma and distress, provoked by dysfunctional affective relationships and, in turn, provoking the overuse of defense mechanisms [33]. This pattern results in somatic symptoms and avoidant responses.

Here, specific features in the linguistic profiles of people with high alexithymia were identified and confirmed by machine learning approach. People with high alexithymia produced more language distortions, used the present tense of auxiliary verbs more frequently, and used fewer possessive determiners than people with low alexithymia. Interestingly, the speech patterns of the former group were less connected. Function words and conjunctions (e.g., articles, adverbs, and determiners) are automatically generated by language systems and serve to map relationships among the consciously generated meaning words (e.g., nouns or verbs) that carry the primary semantic information [34]. In fact, variations in the use of function words provide sensitive indicators of emotional responses and predict the activation of automatic threat detection/response systems better than conventional self-report measures of stress, depression, and anxiety, independent of demographic (age, sex, race), behavioral (smoking, body mass index), and immunological (leukocyte subset distributions) variables [35].

People with high alexithymia used few emotional keywords and many filler words, and the latent semantic structure of their speech was rarely associated with emotion-related words and frequently associated with body-related words, exhibiting a reduced semantic space of emotion. In fact, these participants were more neutral in the continuum of emotional valence and used fewer words associated with anger and disgust. Speaking about specific emotions requires those emotions to be simulated, or partially re-experienced, which involves the same neural and bodily mechanisms recruited during emotional experience [36]. In this framework, alexithymia may be described as difficulty in elevating emotions from the sensorimotor to the symbolic level and in dissociating internal physiological arousal from cognitive appraisal [11,37,38]. Furthermore, our findings suggest that the experiential aspects of emotion schemata are less differentiated in people with high alexithymia than in people with low alexithymia, and our findings also support the idea that alexithymia entails difficulty in emotion conceptualization.

Regarding the acoustic analyses, prosodic and spectral features of language accurately predicted the levels of alexithymia. In fact, variations in voice harmonicity, quality, and signal, three features of speech that are reportedly associated with arousal [39], were negatively associated with alexithymia levels. Interestingly, the microprosodic variation in the length of the fundamental frequency for harmonic sounds, which indicates irregularity in voice intensity and is usually associated with psychological distress [39], was positively associated with alexithymia levels.

Innovatively, this study aggregated linguistic data not only with demographic and psychological data but also with neuroimaging data for its machine learning analysis. We found that cortical thickness at the left temporal pole (positively) and at the isthmus of the right cingulate cortex (negatively) predicted alexithymia levels. In people with high levels of alexithymia, structural and functional alterations are found in the frontotemporal structures, amygdala, insula, cingulate cortex, fusiform gyrus, parahippocampal gyrus, and cerebellum, which are brain areas associated with emotional processing [11,40]. Specifically, emotional processing begins with the identification of emotional stimuli, a function that is mediated by the amygdala and insula [5]. The insula and the cingulate cortex, particularly the posterior subdivision and the right side, merge the internal sensations with cognitive appraisal, resulting in somatosensory awareness of feelings and affective arousal [41]. Cognitive appraisal involves the frontal regions for emotional control and, more importantly, the temporal pole for linkage of interoceptive stimuli with their conceptual meaning [42]. It has been supposed, with good reason, that alterations in the frontotemporal cortex represent the underpinnings of an impaired ability to represent emotions symbolically and linguistically [37]. Notably, Satpute et al. [43] observed that this area was related to semantics and played functionally dissociable roles during emotional experience. Temporoparietal areas had increased activity when individuals retrieved mental state categories relevant for making meaning of their body states.

Starting from the finding that the brain regions involved in semantics are also activated by emotional experience and perception [1,36], we propose that, in people with high levels of alexithymia, altered perception of visceral and somatomotor responses linked to emotional activation may be grounded in alterations in the isthmus of the right cingulate cortex. These alterations could result in impaired emotional awareness and lead to a failure to express emotions verbally and symbolically, grounded in alterations of the left temporal pole. Impaired interoceptive accuracy could result in a dissociation of somatic sensation from semantic knowledge, even to the point of “alexisomia” (difficulty in reading somatic symptoms), an important variable in psychosomatic disorders [37]. Our structural measures and their laterality fit the conceptualization of the left temporal lobe as the neural basis of combinatory syntax and semantics [44] and the right hemisphere as the substratum of emotional elaboration and referential processing [4].

Cutting-edge machine learning analysis of neuro-linguistic data is a promising tool to acquire a deep understanding of the capacity to translate emotional experiences into verbal forms and identify the brain structural correlates of this capacity. This knowledge may have an important impact on health care and “precision psychology”, in which the use of patterns of natural language may provide a useful indicator of implicit well-being.

## 5. Conclusions

In order to obtain a holistic representation of the elements that contribute to the expression of emotional experience, in the present research a forward-looking machine learning methodology combined cross-disciplinary (linguistic, acoustic, neuroimaging, and psychological) outcomes to reveal: -the relationships among the capacity to identify, describe, and regulate emotions; -the ability to translate emotional experiences into verbal forms and sounds; -the cortical structural underpinnings. Through such a methodology, we ascertained the neuro-linguistic pattern of the expression of emotional experience.

## Figures and Tables

**Figure 1 behavsci-12-00292-f001:**
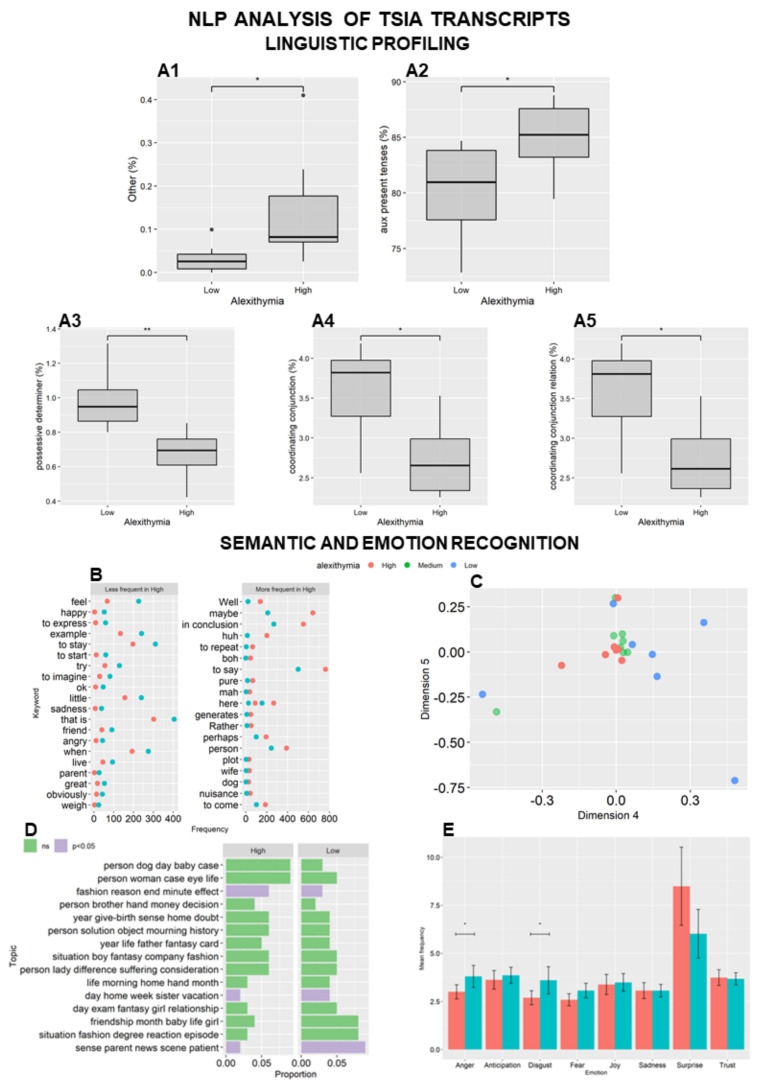
Verbal analysis of Toronto Structured Interview for Alexithymia (TSIA) transcripts. In the Natural Language Processing (NLP), the Linguistic Profiling Analysis showed High Alexithymia subjects produced more pseudowords or non-words (tag: Other) (**A1**), used the present tense of auxiliary verbs more (**A2**), used the possessive determiners less (**A3**), used fewer coordinating conjunctions (**A4**), and less coordinating conjunction relation (**A5**), when compared to Low Alexithymia subjects. In the boxplot are reported the percentages of frequency (median ± percentile). ** P adjusted = 0.01; * at least P adjusted = 0.03. Semantic analysis showed that High Alexithymia subjects used less emotional keywords (feel, happy, to express) and more filler words (maybe, in conclusion, huh) (**B**). This pattern of differential use of words is confirmed by unsupervised Latent Semantic Analysis (LSA), showing cluster separation between High, Medium (participants with middle alexithymia levels), and Low Alexithymia subjects on a two-dimensional space representing the latent semantic structure. The orthogonal dimensions represent semantically related cluster of words, and Low and High Alexithymia subjects are divided by Dimension 4 and 5 in the LSA space (**C**). Unsupervised Latent Dirichlet Allocation (LDA) identified 20 topics, described by the top five words for frequency of use. Three topics were significantly different between High and Low Alexithymia subjects (P ≤ 0.05; Topic 1: fashion, reason, end, minute, effect; Topic 2: day, home, week, sister, vacation; Topic 3: sense, parent, news, scene, patient) (**D**). Exploring sentiment polarization, High Alexithymia subjects were characterized by a reduced frequency of emotion-related words, when compared to Low Alexithymia subjects, specifically for anger and disgust (* at least P = 0.025) (**E**). In the histograms are reported the frequency mean (mean ± standard deviation).

**Figure 2 behavsci-12-00292-f002:**
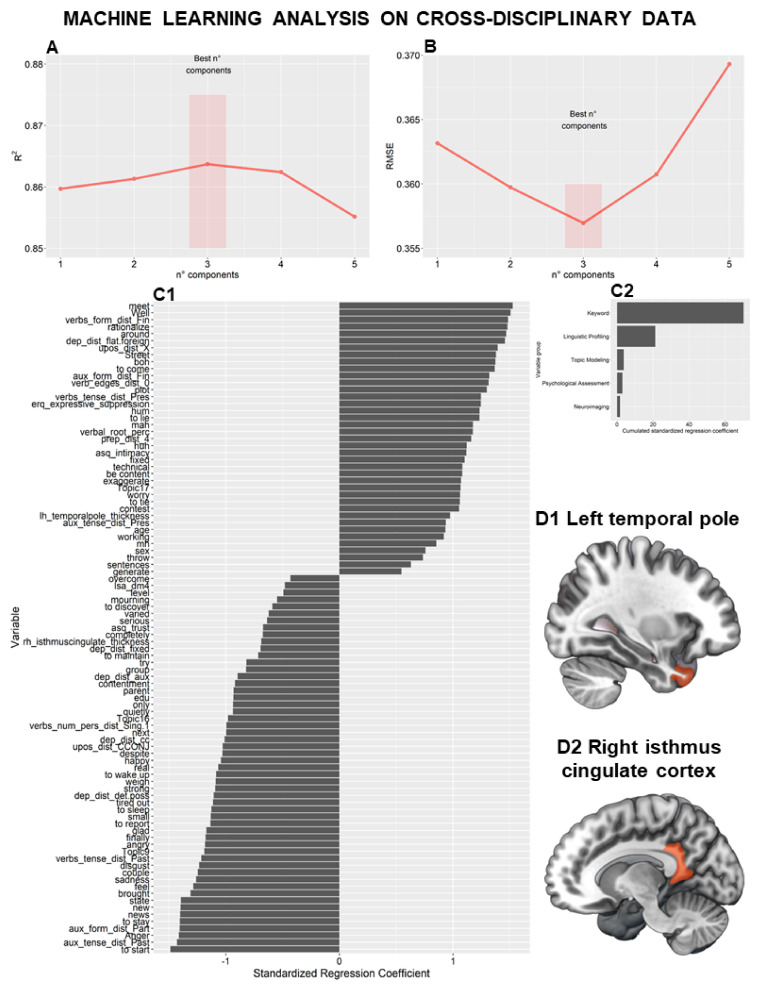
Machine Learning Analysis on Cross-Disciplinary Data. Natural Language Processing (NLP) data were aggregated with demographic, neuroimaging, and psychological data to predict Toronto Structured Interview for Alexithymia (TSIA) scores. Candidate features (n = 92) were extracted using univariate filtering with Spearman correlation method, Principal Components Regression (PCR) model with three components was selected as it had the best performance over evaluation metrics assessed in Leave-One-Out Cross-Validation (LOOCV) (R^2^ = 0.86; Root Mean Square Error—RMSE = 0.35) (**A**,**B**). Variable importance is reported (**C1**,**C2**). Namely, 85 candidate features resulted to belong to NLP, 3 to psychological measures, 2 to neuroimaging data, and 2 to demographic data. The variable importance of neuroimaging data showed that the increased left temporal pole thickness (orange colored area, **D1**) predicted increased TSIA scores, while increased right isthmus cingulate thickness (orange colored area, **D2**) predicted decreased TSIA scores. In (**D1**,**D2**) left is left.

**Figure 3 behavsci-12-00292-f003:**
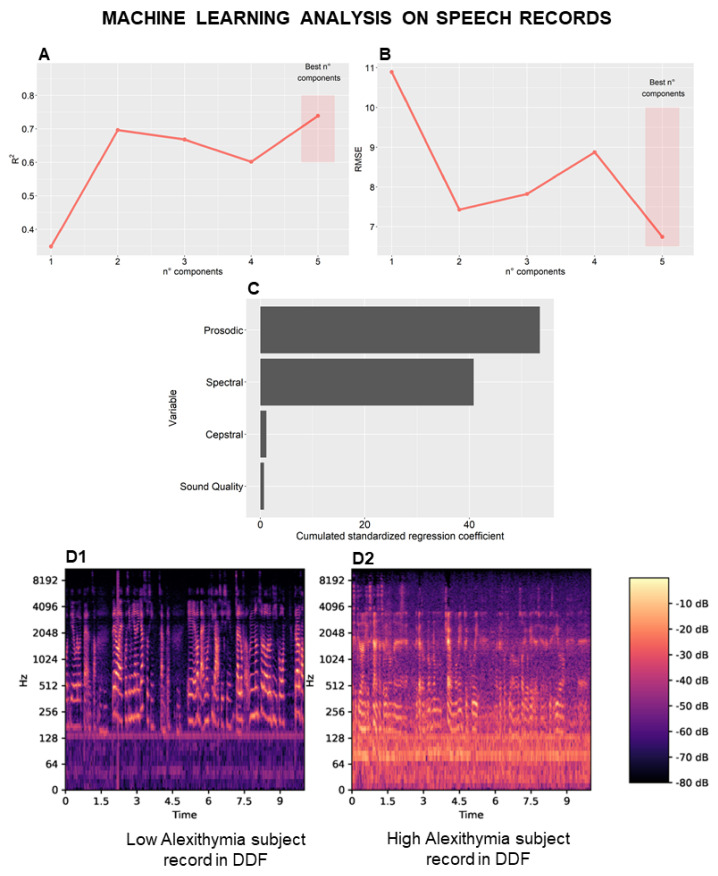
Machine Learning Analysis on Speech Records. Univariate correlation filtering was applied to select 212 candidate acoustic features belonging to the ComParE feature set and Principal Components Regression (PCR) model with 5 components was selected as the best fitting based on R^2^ and Root Mean Square Error (RMSE) metrics in Leave-One-Out Cross-Validation (LOOCV) (R^2^ = 0.74; RMSE = 6.74) (**A**,**B**). Relative variable importance was computed over the domains of ComParE low-level descriptors, namely sound quality, spectral, cepstral, and prosodic (**C**). Representative differences in audio tracks (recorded in responding to an item of Difficulty in Describing Feelings—DDF) from High Alexithymia subjects and Low Alexithymia subjects can be observed in spectrograms (**D1**,**D2**).

**Table 1 behavsci-12-00292-t001:** Sociodemographic variables.

	TSIA DIF	TSIA DDF	TSIA EOT	TSIA IP	TSIA TOT
**Age**	r = 0.30 P = 0.37	r = 0.32 P = 0.48	r = 0.44 P = 0. 17	r = 0.31 P = 0.37	r = 0.41 P = 0.23
**Years of formal education**	r = −0.39 P = 0.23	r = −0.47 P = 0.12	r = −0.36 P = 0.27	r = −0.30 P = 0.37	r = −0.46 P = 0.15

## Data Availability

Not applicable.

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
