# Peer review of "Only Words Count; the Rest Is Mere Chattering: A Cross-Disciplinary Approach to the Verbal Expression of Emotional Experience"

_behavsci, 2022, doi:10.3390/bs12080292_

Round 1

Reviewer 1 Report

I have read carefully the paper titled “Only words count; the rest is mere chattering. A cross-disciplinary approach to the verbal expression of emotional experience”. The aim of the study was to identify neuro-psycho-linguistic pattern of the expression of the emotional experience. To do so the authors assessed neurological, psychological and linguistic pattern that could predict alexithymia levels using a machine learning algorithm.  The topic is interesting and I think that the findings of this research fill the need to explain the relation between cross disciplinary data.

The paper is well written and I really appreciated the Ionesco quote. I suggest a few minor revisions  

Transparency and openness paragraph

-  In this paragraph authors explain a “a priori” analysis of statistical power. The authors did not explain where they found data about mean and standard deviation and effect size used for the analyses.  

- Authors state that analyses were carried out on three groups (LA, MA, HA) in lines 119-120 they have reported only the data regarding LA and HA, what about MA group? Also, if there are multiple comparison between groups I think that a p-value correction for multiple comparisons is needed.

-  In the power analysis for the Wilcoxon test authors used F value, why this choice? Wilcoxon effect size should be indicated as “d”

- In Lines 118 - 119 authors specify in parentheses Mean standard deviations of the LA group, I think the acronym of one group (HA) was missed.  

Machine learning paragraph

-Line 249- 269 Authors divided dataset according to NLP, neuropsychological, neuroimaging and demographic variables and according to audio features of speech, I think that authors should explain why they made this choice

- Authors used the LOOCV algorithm to test the PCR model, they explained that LOOCV uses n-1 samples for training, for results interpretation it is necessary to know the n value

Author Response

Answers to Reviewer 1

-  Transparency and openness paragraph: in this paragraph authors explain a “a priori” analysis of statistical power. The authors did not explain where they found data about mean and standard deviation and effect size used for the analyses. 

The data used in the a priori power analysis was retrieved from our LA and HA samples and this has been clarified in the manuscript at lines 121-122.

To further detail such an issue, we would like to add some additional analyses supporting our choice. As the HA and LA groups are identified based on the TSIA score, which requires a trained psychologist to be administered and scored, it is difficult to find similar data in literature. In this kind of situation, it’s common practice to rely on a compromise power analysis, where both α and 1 − β are computed as functions of effect size, N, and an error probability ratio q = β/α. Here, we used q = 1 to balance error risks and we obtained α = 0.05 and statistical power = 0.94 with nHA= 7 and nLA = 7. We believe that our sampling choice does not bias the results as the a priori power analysis defines required sample size as a function of power level 1 − β, significance level α, and the to-be detected population effect size that is computed from mean and standard deviation provided, so the analysis was blind to sample size information. In fact, this result is stable even in a post hoc power analysis, where we confirmed a statistical power (1 − β) of 0.94. Moreover, when a sensitivity power analysis is performed, we obtain a population effect size equal to 1.91, which is roughly similar to the one we obtained plugging in our data in the a priori power analysis (1.87). In this sensitivity analysis the population effect size is defined as a function of α, 1 − β, and N (where n is our sample size for LA and HA) so we are advancing that the results we obtained in the a priori power analysis are stable.

- Authors state that analyses were carried out on three groups (LA, MA, HA) in lines 119-120 they have reported only the data regarding LA and HA, what about MA group? Also, if there are multiple comparison between groups I think that a p-value correction for multiple comparisons is needed.

The comparisons of HA and LA subjects with MA subjects were reported in the original version in the Supplementary Materials, as specified at lines 312, 326, 356, 367. Namely, Natural Language Processing results about MA comparisons are listed in the supplementary results as well as in the supplementary Figures 1, 2 and 3, while Machine Learning results are reported in supplementary Table 2A and B. The p-value correction was performed by computing the False Discovery Rate when multiple tests occurred within a comparison. In fact, there is an FDR p-adjusted column in each result table for both main and supplementary results. As our experimental design aimed at comparing HA and LA subjects, we did not perform p-value correction for cross-comparison tests. We reported MA vs HA and MA vs LA results for the sake of completeness (lines 312-313) and we were not aiming to find false positive results accounting for comparisons out of the scope of our experimental design.

-  In the power analysis for the Wilcoxon test authors used F value, why this choice? Wilcoxon effect size should be indicated as “d”

We thank the reviewer for this observation and we do agree that effect size in Wilcoxon Testing should be indicated as “d”. We corrected this typo at line 121.

- In Lines 118 - 119 authors specify in parentheses Mean standard deviations of the LA group, I think the acronym of one group (HA) was missed. 

Thank you, we corrected this error in the same lines.

Machine learning paragraph

- Line 249- 269 Authors divided dataset according to NLP, neuropsychological, neuroimaging and demographic variables and according to audio features of speech, I think that authors should explain why they made this choice

Thank you for your question. We decided to divide our multi-source data in two datasets for two main reasons, the first being avoiding overfitting, as the audio features dataset contained more than 200 features after feature selection (line 394) and the other dataset contained 92 features after feature selection (line 375). We thought that combining these features could inflate the performance metrics. The second reason is that, to the best of our knowledge, this was the first time ever that audio features were extracted from an alexithymia speech sample, so we wanted to be able to discuss the results from a dataset independently from the others. In the case of audio features, we had a specific experimental question: can the acoustic features of free speech alone be used to predict the TSIA score? To answer this question, we needed to build an ML model with acoustic features. We updated the manuscript with this information (lines 280-282).

- Authors used the LOOCV algorithm to test the PCR model, they explained that LOOCV uses n-1 samples for training, for results interpretation it is necessary to know the n value

Thank you for your question. We specified the n value in the manuscript (line 267). We had n = 14 (nHA = 7, nLA = 7).

Reviewer 2 Report

This was a very interesting report of the brain correlates of the capacity to translate emotional experience into words which were determined through cortical thickness measures by means of MRI. I thought the applying of a specific methodology, combining a Machine Learning methodology with analyzing transcripts and speech recordings obtained through the Toronto Structured Interview for Alexithymia, was particularly interesting.

Along with collecting neuroimaging, linguistic and demographic data, the participants underwent an extensive psychological assessment, including many evaluations, among them attachment styles, emotional regulation and coping strategies, depression and anxiety levels.

The authors claim that they identified the neuro-psycho-linguistic pattern of the expression of emotional experience.

While in principle I find the study very interesting with the potential to be theoretically informative, I believe the report, as it currently stands, has some serious limitations and requires substantial revision.

1.     When describing the study participants, their average age and gender are not indicated, moreover, the number of participants is not indicated either in the abstract or in the appropriate section of the article (lines 91-92). And only from the section 2.3. we learn that 22 subjects took part in the study.

2.     The project sample is very small to study such highly variable indicators as personality traits. At the same time, the participants differ in such important indicators as gender, age, education.

3.     It is unclear how exactly were the samples LA (n = 7), MA (n=8), HA (n=7) formed? How were the participants distributed in these 3 samples by gender, age, education?

4.     Table 1 shows 5x16=80 calculated correlations, 80-fold verification of statistical significance. Correction for 80-fold checks is required, for example, the Benjamini-Hochberg Procedure.

The numerous correlations between psychological indicators presented in Table 1 may indicate some common reasons for their joint variability, which are not necessarily reduced to alexithymia. For example, education level, gender, etc.

5.     The image quality of Figures should be improved, since in its current form it is not readable.

6.     You wrote: “In accordance with previous findings [11], males showed higher alexithymia levels than females” (lines 440-441). So how many participants were males, and how many females of those 22?

Author Response

Answers to Reviewer 2

- When describing the study participants, their average age and gender are not indicated, moreover, the number of participants is not indicated either in the abstract or in the appropriate section of the article (lines 91-92). And only from the section 2.3. we learn that 22 subjects took part in the study.

The Reviewer 2 was right. In the revised version we have added the required information in the Abstract and Materials and Methods (lines: 92-93).

- The project sample is very small to study such highly variable indicators as personality traits.

The Reviewer 2 was right. However, the sample size for hypothesis testing was computed using an a priori power analysis for the Wilcoxon rank-sum test for two independent groups, performed on one of the most statistically significant parameters from linguistic profiling analysis. The total sample size required was estimated in 14 subjects (7/group), with É‘ err prob=0.05, Effect size f=1.92, achieved statistical power=0.95.

Furthermore, we decided to apply a robust data analysis workflow to provide replicable and reliable results. To avoid overfitting due to sample size and high dimensionality in our data we performed a two-stage feature selection process including dimensionality reduction, allowing us to summarize variance in the data in a few features (5 or 3 components), thus counteracting potential biases. Results from Machine Learning modeling were validated with the leave-one-out cross-validation technique, which allowed us to better assess generalizability while producing stable findings. To ensure results replicability, a seed was set to 12345 as guidance for algorithms needing pseudorandomization and R code is available upon reasonable request.

- At the same time, the participants differ in such important indicators as gender, age, education…and… How were the participants distributed in these 3 samples by gender, age, education?

Here we reported the first lines of the Supplementary Table 1.

Total
Sample

High
Alexithymia

Medium
Alexithymia

Low
Alexithymia

Number

(Males)

22

(12)

7

(6)

8

(4)

7

(2)

Age

Mean in years

(±SD)

44.32

(13.17)

52.14

(4.98)

41.88

(12.90)

39.29

(16.70)

Years of formal education

Mean in year (±SD)

15.68

(2.40)

15.00

(1.73)

15.13

(3.23)

17.00

(1.41)

Following the Reviewer’s criticism, we performed additional analyses and we found that LA, MA, and HA subjects do not differ in age (Kruskal-Wallis test P=0.37) or years of formal education (Kruskal-Wallis test P=0.36). We have reported the results at lines 299-300.

- It is unclear how exactly were the samples LA (n = 7), MA (n=8), HA (n=7) formed?

We specified how samples were divided in LA, MA and HA based on the 0.25 and the 0.75 quantiles of the TSIA score distribution at line 117-118.

- Table 1 shows 5x16=80 calculated correlations, 80-fold verification of statistical significance. Correction for 80-fold checks is required, for example, the Benjamini-Hochberg Procedure.

The numerous correlations between psychological indicators presented in Table 1 may indicate some common reasons for their joint variability, which are not necessarily reduced to alexithymia. For example, education level, gender, etc.

Following the Reviewer’s suggestion, we corrected the correlations for multiple testing using False Discovery Rate (Benjamini-Hochberg Procedure) and updated the results accordingly.

- The image quality of Figures should be improved, since in its current form it is not readable.

Thank you for the suggestion, we improved the figure quality.

- You wrote: “In accordance with previous findings [11], males showed higher alexithymia levels than females” (lines 440-441). So how many participants were males, and how many females of those 22?

In the revised version we have added this information in the Materials and Methods (lines: 92-93). Furthermore, in the present and previous version of the manuscript this information was reported in the Supplementary Table 1.

Round 2

Reviewer 2 Report

Dear authors you will find my comments and suggestions in the attached file.

Author Response

Well, I’ve realized that this is convincing for linguistic indicators. But for psychological indicators this is more than doubtful… It is very good that you have updated the results. But the main problem has not been eliminated. Table 1 is not informative, especially considering the number of the participants in the samples. I think that the inclusion of psychological indicators in this study in its current form significantly reduces the quality of the publication. The psychological interpretations given as they are now do not correspond to the desired level. My suggestion is to exclude psychological indicators from the publication. At least, what is presented in Table 1.

To meet the Reviewer’s criticism, we removed phycological indices and their relation with TSIA scores, also modifying the Table 1. We have included the psycological variables only in the Machine Learning analysis of cross-disciplinary data in which a robust data analysis workflow was to applied to provide replicable, reliable, and validated results.

Such a small amount of the participants in your sample will inevitably show the insignificance of differences in such important indicators as gender and education.

The Reviewer2 was right. To meet her/his criticism, we added this limitation reganding the reported not significant results.

The question is not how you divided these 3 samples, but how many subjects were initially surveyed. From which initial population were those 22 participants selected? From the description of the sample, it is not clear if it was randomized.

We apologize to the Reviewer2 for not having well understood her/his previous question. A sample of 79 healthy subjects (36 males; mean age ± SD: 40.06 ± 12.57 years; range: 19-59; Males: 38.13 y ± 12.24; Females: 41.67 y ± 12.76) belonging to a larger group of healthy volunteers (N = 125), submitted to Magnetic Resonance Imaging scan protocol for other studies, were enrolled in the present research. Only those who accepted to come again to Santa Lucia Foundation were tested on TSIA. Starting from the a priori power analysis to compute the sample size, we included in the present study 22 subjects (LA: n = 7, MA: n=8, HA: n=7). Educational level ranged from an eighth grade to a post-graduate degree (mean education years ± SD: 15.83 ± 2.86; range: 8-25). All participants were right-handed as assessed with the Edinburgh Handedness Inventory. Inclusion and exclusion criteria are reported in the main text. We have reported this information in the revised version of the manuscript.